# Restriction of Flaviviruses by an Interferon-Stimulated Gene SHFL/C19orf66

**DOI:** 10.3390/ijms232012619

**Published:** 2022-10-20

**Authors:** Youichi Suzuki, Takeshi Murakawa

**Affiliations:** 1Department of Microbiology and Infection Control, Faculty of Medicine, Osaka Medical and Pharmaceutical University, 2-7 Daigaku-machi, Takatsuki 569-8686, Japan; 2Department of Biochemistry, Faculty of Medicine, Osaka Medical and Pharmaceutical University, 2-7 Daigaku-machi, Takatsuki 569-8686, Japan

**Keywords:** SHFL, C19orf66, flavivirus, interferon-stimulated antiviral gene, AlphaFold2

## Abstract

Flaviviruses (the genus *Flavivirus* of the *Flaviviridae* family) include many arthropod-borne viruses, often causing life-threatening diseases in humans, such as hemorrhaging and encephalitis. Although the flaviviruses have a significant clinical impact, it has become apparent that flavivirus replication is restricted by cellular factors induced by the interferon (IFN) response, which are called IFN-stimulated genes (ISGs). SHFL (shiftless antiviral inhibitor of ribosomal frameshifting) is a novel ISG that inhibits dengue virus (DENV), West Nile virus (WNV), Zika virus (ZIKV), and Japanese encephalitis virus (JEV) infections. Interestingly, SHFL functions as a broad-spectrum antiviral factor exhibiting suppressive activity against various types of RNA and DNA viruses. In this review, we summarize the current understanding of the molecular mechanisms by which SHFL inhibits flavivirus infection and discuss the molecular basis of the inhibitory mechanism using a predicted tertiary structure of SHFL generated by the program AlphaFold2.

## 1. Introduction

There are many microorganisms, including viruses, in and around us, many of which do not cause us any harm. Some microorganisms, however, disrupt human health and ultimately cause disease through infection. In response, the immune system is activated in the host to counteract the pathogen. The interferon (IFN) system is considered to play a central role in the innate immune response against virus infection [1]. IFN, which is produced by virus-infected cells following the recognition of viral components by cell-encoded pattern-recognition receptors [2], binds to the specific surface receptor (IFNR) of cells in an autocrine or paracrine manner, resulting in the induction of an antiviral state in the cells. Many studies have demonstrated that the IFN-mediated antiviral state is established by the molecular functions of a variety of cellular factors [3]. Recently, it has become apparent that a protein product of the *C19orf66* gene, SHFL (official gene symbol given by the HUGO Gene Nomenclature Committee [4]), is one of the cellular antiviral factors whose expression is upregulated by the IFN response [5,6]. Interestingly, as Rodriguez and Muller comprehensively reviewed in another article [7], SHFL exhibits antiviral activity against various types of RNA and DNA viruses with various mechanisms of action, though the molecular mechanisms are not entirely clear. This review will focus on SHFL-mediated inhibition of flavivirus replication and discuss the molecular and structural process by which SHFL acts as an anti-flaviviral factor.

## 2. Flaviviruses

The genus *Flavivirus* (hereafter referred to as “flavivirus”) is a group of enveloped and positive-sense RNA viruses belonging to the *Flaviviridae* family. The *Flaviviridae* family is also composed of three other genera, *Hepacivirus*, *Pegivirus*, and *Pestivirus* [8]. An important characteristic of the *Flaviviruses* is that this genus consists of more than 50 species, most of which are transmitted by arthropods such as mosquitoes and ticks. Particularly, some of the flaviviruses are clinically significant, since they can potentially cause hemorrhaging (dengue virus [DENV] and yellow fever virus [YFV]) or encephalitis (West Nile virus [WNV], tick-borne encephalitis virus [TBEV], Japanese encephalitis virus [JEV]) in humans [9]. Additionally, the Zika virus (ZIKV), a flavivirus family that spread explosively throughout the Americas in 2015, has been shown to be associated with Guillain–Barré syndrome and neurological complications [10].

The flavivirus virion contains a viral genome comprising an ~11-kb-long single-stranded RNA, whose 5′-end is capped with m7GpppAmp and 3′-end is not polyadenylated (Figure 1A). The binding of the flavivirus to surface entry receptors such as heparan sulfate, C-type lectins, and phosphatidylserine receptors results in internalization of the attached virion via clathrin-dependent endocytosis (Figure 1B) [11]. After the release of the viral genome into the cytoplasm, a single long open reading frame (ORF) in the viral RNA, encoding three structural (capsid [C], pre-membrane [prM], and envelope [E]) and seven non-structural (NS1, NS2A, NS2B, NS3, NS4A, NS4B, and NS5) proteins (Figure 1A), is translated, and the polyprotein is subsequently processed by cellular and viral proteases. Serine protease activity embedded in the N-terminal domain of NS3 is responsible for the polyprotein processing [12]. New copies of the viral RNA are synthesized by the RNA-dependent RNA polymerase (RdRp) activity of NS5 from the intermediate negative-sense RNAs that are generated from the input positive-sense RNA genome. Replication of the viral RNA occurs in the unique membranous structures formed in the endoplasmic reticulum (ER), which are considered to be a microenvironment that provides factors necessary for the activity of the replication complex [9]. The assembly of nucleocapsids (a viral RNA genome packaged by the C proteins) with prM and E proteins to form immature particles also occurs within the ER lumen, and the nascent virions undergo the maturation process through the furin-mediated cleavage of prM and conformational changes in the E glycoproteins in the trans-Golgi network [13]. Eventually, mature infectious virions’ egress from the infected cell occurs through exocytosis (Figure 1B).

## 3. Identification of SHFL as a Novel Cellular Inhibitor against Flaviviruses

There are three different classes of IFN, types I, II, and III, in mammals. Although type-I (the largest IFN family, consisting of IFN-α, -β, -ε, -κ, and -ω) and type-III (only consisting of IFN-λ) IFNs are considered to play major roles in the induction of an antiviral state, type-II IFNs including IFN-γ, is also a potent inducer of antiviral responses [1]. Type-I and -III IFNs that bind to cognate IFNR activate the cytoplasmic JAK-STAT signaling pathway, resulting in the formation of IFN-stimulated gene factor 3 (ISGF3), whereas type-II IFN activates the JAK–STAT pathway through a specific receptor complex called IFNGR and triggers the formation of the IFN-γ activation factor (GAF). ISGF3 and GAF act as transcription factors for the expression of hundreds of genes via binding to regulatory elements on the chromosomes (reviewed in [1]).

The cellular genes induced during the IFN response are defined as IFN-stimulated genes (ISGs), and their many gene products are thought to function as antiviral factors [3]. SHFL was first listed as an antiviral ISG candidate in a comprehensive overexpression screen to search for inhibitory genes against hepatitis C virus (HCV), a *Hepacivirus* [14]. Schoggins et al. generated a set of lentiviral vectors expressing more than 380 human ISGs and reported that SHFL (shown as FLJ11286) was listed in the gene-set, potentially inhibiting GFP-expressing HCV replication in Huh-7.5 cells [14].

Subsequently, SHFL was also identified as an antiviral ISG against DENV, a mosquito-borne virus causing dengue fever in humans [15]. In the study, a gain-of-function cDNA-expression cloning approach [16] was used to identify cellular genes suppressive of DENV, though the cDNA library was generated with the mRNA of HeLa cells treated with type-I IFN. Whereas Huh-7.5 cells exhibited a massive cytopathic effect during DENV infection, the transduction of the cDNA library by lentiviral vectors conferred resistance to DENV-induced cell death. Sequencing analysis using a BLAST search showed that SHFL cDNA was found to be introduced in more than 50% of surviving Huh-7.5 cell clones [15]. The anti-DENV activity of SHFL was confirmed via overexpression experiments, in which the replication of all DENV serotypes (DENV-1 to 4) was inhibited by SHFL expression. Importantly, the endogenous expression of SHFL was upregulated by type-I, -II, and -III IFN treatments, and when endogenous SHFL expression was depleted by RNA interference (RNAi), the suppressive activity of IFNs against DENV became less effective. These results indicated that SHFL was critical to the IFN-mediated anti-DENV response in human cells. In contrast, the expression of SHFL had no influence on the activation of IFN-β or other ISGs in human cells, indicating that SHFL is not a regulator in the IFN signaling pathway [15]. Notably, SHFL (i.e., *C19orf66*) is named “repressor of yield of DENV” (RyDEN) for the first time in this study, as this gene had not previously been given a special nomenclature [15].

## 4. Domain Features of SHFL

SHFL is a 291-amino-acid protein encoded in an 8-exon gene located on genomic region 19p13.2. [15]. Although a shorter isoform lacking amino acids 164–199 was also registered in the NCBI database [17,18], it remains unclear whether the shorter version of SHFL is indeed expressed in human cells. The initial prediction of a secondary structure using the JPred program [19] showed that SHFL consisted of eight α-helixes and seven β-strands [15]. However, a recent prediction of the tertiary structure of SHFL generated by the program AlphaFold2 [7,20,21] reveals that the seven N-terminal α-helixes (α1 to α7, Figure 2) form a semi-globular structure (referred to hereafter as the helix-rich domain) and the characteristic glutamate (E)-rich domain in the C-terminus forms an elongated α-helix (α10, Figure 2 and Figure 3A). Moreover, SHFL possesses another domain comprised of nine β-strands (β1 to β9, Figure 2) and loops (referred to hereafter as the flexible domain) on the opposite side of the helix-rich N-terminal domain (Figure 3A). These two domains may construct a large flexible cavity accessible to co-factors of SHFL (Figure 3A) [7]. Although, in a comparison of amino acid sequences, mammalian proteins showing a high similarity to SHFL were not found, a putative nuclear localization signal (NLS) and nuclear export signal (NES) have been predicted to be present in the middle (121–137) and C-terminal (261–269) regions, respectively (Figure 2) [15]. Indeed, SHFL has been shown to be mainly localized in the cytoplasm [15,22,23,24]. However, when the C-terminal domain containing an NES (251–291) was deleted, the SHFL deletion mutant accumulated primarily in the nucleus [15], suggesting that SHFL is basically a nucleocytoplasmic shuttling protein that is retained in the cytoplasm at steady state. Therefore, SHFL may have different functions in the cytoplasm and nucleus.

## 5. Molecular Aspects of SHFL in the Inhibition of Flaviviruses

In our study using a gain-of-function screen of a cDNA library derived from IFN-treated human cells, SHFL was first identified as a cellular factor restricting DENV replication. DENV is classified into four serotypes (DENV-1 to 4) based on the amino acid sequences of the E protein that determine the sensitivity to neutralizing antibodies [25]. The overexpression of SHFL conferred resistance to all DENV serotypes in human hepatoma cell lines. When the endogenous expression of SHFL was reduced through RNA interference, the suppressive activity of type-I IFN against DENV became less effective, suggesting that SHFL was one of the critical contributors to the IFN-mediated inhibition of DENV replication [15]. In a subsequent report, the enhanced replication capability of DENV by the knockout of SHFL was also shown in a human lung carcinoma cell line [22]. Importantly, infection by other hemorrhagic and encephalitic flaviviruses, including WNV, YFV, JEV, and ZIKV, was found to be inhibited by the expression of SHFL [15,22,23,26,27], suggesting that SHFL is a pan-flaviviral cellular inhibitor.

The question to ponder is: what is the molecular basis for the inhibition of flaviviruses by SHFL? Several models have been proposed by our and other groups for the mechanism involved in SHFL-mediated flavivirus suppression, which can be summarized as (i) translational inhibition, (ii) RNA degradation, (iii) protein degradation, and (iv) frameshift inhibition (Table 1) [7,15,22,23,26].

An affinity-purification mass-spectrometry (MS) analysis revealed that SHFL was associated with two other cellular proteins, poly(A)-binding protein cytoplasmic 1 (PABPC1) and La-motif-related protein 1 (LARP1). PABPC1, a member of the PABP family, bridges the 5′ and 3′ ends of mRNA by binding to both eIF4G and the poly(A) tail, stimulating the initiation of translation [28]. Interestingly, although the DENV RNA genome lacks a terminal poly(A) tail, PABP has been shown to interact with the 3′ untranslated region (UTR) of DENV RNA by recognizing the A-rich regions upstream of the 3′ stem loop in the 3′ UTR [29]. LARP1 is also an RNA-binding protein and has been reported to interact with PABP and eIF4E to facilitate translation initiation [30]. Interestingly, although SHFL exhibits RNA-binding activity, the binding specificity to DENV RNA was enhanced by the presence of PABPC1 in vitro [15]. Considering the role of PABP and LARP1 in the mRNA translation process [31], the complex formation of SHFL with PABPC1 and LARP1 suggests that SHFL interferes with the translation of DENV RNA by inhibiting PABPC1 and LARP1 functions. Supporting this, an RNA interference-mediated depletion of PABPC1 and LARP1 significantly reduced the level of DENV replication, and SHFL also inhibited the protein expression from a replication-defective DENV sub-genomic replicon RNA [15].

An alternative mode of action of SHFL in the inhibition of DENV has been proposed. A co-immunoprecipitation followed by MS analysis performed by Balinsky et al. revealed that SHFL displayed a strong preference for interacting with nucleic-acid-binding proteins [22]. Importantly, MOV10, one of the top MS hits, was found to be co-localized with SHFL at the viral-NS3-positive area in the perinuclear region of the DENV-infected cell, indicating that SHLF and MOV10 associate with the viral replication complex [22,32]. MOV10 is an ATP-dependent RNA helicase, which is reported to participate in cellular antiviral defense by regulating microRNA expression and mRNA stability [33]. Interestingly, in uninfected cells, the co-localization of SHFL and MOV10 was observed in the processing (P) body, which is a cytoplasmic granule involved in mRNA decay and translational suppression [34]. Therefore, it can be proposed that, upon flavivirus infection, SHFL and MOV10 are recruited from the P-body to the region where the flaviviral RNA amplification process takes place and collaborate to induce the destabilization of viral RNA. Similar behavior of the P-body component was also observed with the 5′–3′ exonuclease XRN1 [22,35]. In addition, PABPC1 and LARP1 have been demonstrated to be involved in mRNA decay either in the P-body or stress granules [36,37], though the localization of these SHFL co-factors in virus-infected cells has not been shown.

SHFL has also been suggested to induce the degradation of a critical enzyme of the flavivirus. NS3, a bipartite non-structural protein comprised of N-terminal protease and C-terminal NTPase/helicase domains, plays a central role in viral polyprotein processing and RNA replication processes in the intracellular membrane structures within the ER, where the flavivirus replication complex is formed [12,32,38]. Recently, two studies have reported that ZIKV and JEV replication were suppressed by the expression of SHFL, and interestingly, both studies showed that SHFL reduced the protein level of NS3, which was not attributed to the downregulation of mRNA [23,26]. The reduced expression of NS3 was likely mediated by the lysosomal degradation pathway, since it could be restored by treatment with lysosomal inhibitors (ammonium chloride, chloroquine) but not with a proteasomal (MG132) or autophagy (3-methyladenine) inhibitor [23,26]. Although the interaction of SHFL and NS3 was also shown, these data were only obtained in the co-transfection experiments using SHFL and NS3 expression plasmid DNA. Therefore, further work is required to determine whether the lysosomal degradation of NS3 indeed occurs in virus-infected cells and is a primary or additional mechanism of the SHFL-mediated inhibition of flavivirus infection. Regardless, NS3 appeared to be a bona fide interactor with SHFL in DENV-infected cells and was not influenced by RNase treatment (i.e., protein–protein interaction) [22]. Since DENV NS3 is reported to circumvent the host antiviral response through the antagonization of critical molecules in innate immunity such as STING and 14-3-3 proteins [39,40,41], the possibility that SHFL induces the degradation of NS3 is an interesting scenario from the perspective of an arms race waged between the host defense system and the flavivirus [42].

In a study reporting the SHFL-mediated destabilization of JEV NS3, an additional inhibitory action of SHFL was shown, potentially restricting the programmed ribosomal frameshifting (PRF) that occurs during JEV gene expression [26]. PRF is an alternative translation mechanism, found mostly in viruses, which enables the expression of overlapping ORFs in a controlled manner [43]. One of the well-characterized PRFs is -1PRF, used in human immunodeficiency virus type 1 (HIV-1) Gag–Pol fusion protein expression, where the ribosome slides back one nucleotide to the 5′ direction at the end of the Gag ORF so that the Pol ORF becomes in-frame with the Gag reading frame [43,44]. Importantly, SHFL has been demonstrated to act as an IFN-inducible inhibitor against the -1PRF of HIV-1 [18]. In the case of JEV, the -1PRF is predicted to occur at the 5′ end of NS2A, resulting in the production of a longer version of the NS1 protein termed NS1′ [45], and the expression of NS1′ was shown to be diminished by SHFL [26]. Although the precise role of NS1′ in the JEV replication cycle remains unclear, it has been reported that this NS1 derivative is also synthesized from the -1PRF in the infection of another neuropathogenic flavivirus, WNV, and it plays some role in the neuroinvasiveness of WNV in mice [46]. Thus, future studies will clarify whether the inhibition of the -1PRF by SHFL reduces the virulence of neuropathogenic flaviviruses such as JEV and WNV in vivo. It should be noted that SHFL was also shown to inhibit the -1PRF of severe acute respiratory syndrome coronavirus 2 (SARS-CoV-2), but the results were obtained only by using a single-tube biochemical assay in which the expressions of two different reporters separated by the -1PRF element were compared in the presence or absence of SHFL [47,48]. Importantly, it has been demonstrated that the replication of SARS-CoV-2 in cultured cells was not inhibited by the overexpression of SHFL [27]. Therefore, the inhibition of the -1PRF by SHFL may not be a common mechanism to restrict the replication of all viruses with the -1PRF mechanism.

## 6. Structural Interpretation of Key Domains in SHFL—Seeing Is Believing

In the secondary structure of SHFL, the NLS (residues 121–137, Figure 2) overlaps with a domain of residues (112–135) that is predicted to form a zinc finger motif [15,49]. Zinc fingers are small protein domains that are found in a number of proteins involved in various cellular processes and have been shown to function as interaction domains with nucleic acids and proteins [49]. Based on the spatial structure similarity, Krishna et al. classified the zinc finger motif into eight groups, and among them, the zinc finger of SHFL was inferred to be categorized into the zinc ribbon fold group, which contains the CXXC(H)-15/17-CXXC sequence motif [49,50]. In the putative zinc finger motif of SHFL, the prosthetic metal, Zn^2+^, is coordinated by four cysteine residues (Cys112, Cys115, Cys132, and Cys135) by Zn^2+^–S bonds of 2.3–2.4 Å in length (Figure 3B). The zinc ribbon fold is composed of two β-hairpins that form two zinc-binding sub-sites, revealing that SHFL’s zinc ribbon is similar to that of the ribosomal protein L36 of *Thermus thermophilus* [49,50].

Of particular interest is that this zinc ribbon domain has been demonstrated to be critical in executing the anti-flavivirus activities of SHFL. In our initial study, the NLS was shown to be an interaction domain with PABPC1, and the alanine substitution of arginine and lysine residues within the NLS (Arg121, Arg122, Arg126, Lys127, Arg131, Arg133, Lys134, Arg136, and Lys137) reduced the binding efficiency of SHFL to PABPC1. More importantly, the NLS mutant of SHFL lost its inhibitory activity against DENV, indicating that positively charged residues of NLS play an essential role in the inhibition of DENV replication, probably through the interaction with PABPC1 [15]. Additionally, the alanine substitution of four Zn^2+^-coordinating cysteine residues diminished the antiviral effect of SHFL against JEV [26]. Supporting this, Kinast et al. have reported that certain cysteine residues were also important for the restriction capacity against HCV [24]. Significantly, Arg131, Arg133, and Arg 136 interact with Asp271/Glu274, Glu262/Glu270, and Glu277, respectively, in the elongated α-helix (α10) in the C-terminus (Figure 3B), suggesting that the central zinc ribbon domain (colored green in Figure 3A) and the C-terminal E-rich domain (colored magenta in Figure 3A) form a core region via the electrostatic action of the basic (i.e., arginine) and acidic (i.e., glutamic acid and aspartic acid) amino acid residues present in the respective domains. Therefore, disruption of the critical residues in the core region may cause the destabilization of the entire structure of SHFL, resulting in the dysfunction of its antiviral activity. Indeed, our co-immunoprecipitation analysis using N- and C-terminal deletion mutants of SHFL indicated that the interaction with PABPC1 was weakened not only when the central region (residues 102–151) containing the zinc ribbon domain was absent, but also when SHFL lacked the C-terminal region (residues 251–291, [15] and our unpublished data). Interestingly, a biochemical study using a recombinant protein and in vitro-transcribed RNA showed that three arginine residues in the zinc ribbon domain (Arg131, Arg133, and Arg136) were involved in the RNA-binding activity of SHFL. Hence, this core region may serve as a port to allow other interactors (proteins and/or nucleic acids) to be accessible to SHFL.

Meanwhile, another domain of SHFL was also shown to be involved in suppressing the viral -1PRF mechanism. Wang et al. demonstrated that even when a predictable splicing variant of SHFL lacking 36 amino acids (residues 164–199) was overexpressed, the short form of SHFL had little effect on the inhibition of -1PRF in HIV-1 Gag–Pol expression [18]. As with HIV-1, the -1PRF of JEV was not suppressed by the expression of the SHFL short form, although mutant SHFL continued to exhibit weaker but significant inhibitory activity against JEV infection [26]. From the structural point of view, the region between residues 164 and 199 composes a large part of the flexible domain (colored orange in Figure 3A). Notably, the loop formed in the 36-amino-acid region is likely to associate with the loop region immediately before the C-terminal α10 via a hydrophobic interaction (Figure 3C). Thus, the lack of the 36-amino-acid region may also result in a disorder of the overall integrity of SHFL. However, future structural studies will be required to clarify how these different domains are coordinated in the execution of inhibitory activity against various types of viruses, and their interactions with proteins and nucleic acids.

## 7. Anti-Flavivirus Activity of SHFL In Vivo

Although the inhibitory activity of SHFL against flaviviruses has been well demonstrated in cell-based in vitro experiments, one more important question pertains to whether SHFL contributes to the anti-flaviviral immune defense in vivo. The answer is likely yes; a recent study by Hanners et al. generated SHFL knockout (KO) mice and evaluated ZIKV replication and pathogenesis in the KO mice [27]. The results showed an apparent increase in viral RNA in the liver and spleen of SHFL KO mice compared to wild-type (WT) mice, and it is noteworthy that the clinical symptoms induced by the ZIKV infection were more severe in the KO mice. Moreover, the levels of ZIKV titer in the brain and spinal cord of KO mice were significantly higher than in WT mice, which was reflected by increased inflammation in the central nervous system [27]. These data demonstrate that SHFL plays a critical role in the host defense against ZIKV infection and pathogenesis (Table 1). The KO mice would be a powerful model to analyze the inhibitory role of SHFL in other flavivirus infections in vivo.

## 8. Conclusions

Broad-spectrum antiviral activity is undoubtedly a unique feature of SHFL. In this review, we focused only on the inhibitory activity of SHFL against flaviviruses, as the antiviral function of SHFL was first demonstrated in studies of *Flaviviridae* family viruses (i.e., HCV and DENV studies [14,15,22]). However, as Rodriguez and Muller well summarized in their review [7], the virus inhibitory activity of SHFL is not limited to flaviviruses but extends to many other types of RNA and DNA viruses, and interestingly, various inhibitory mechanisms of action have been proposed for SHFL’s antiviral activities [15,18,22,23,24,26,27,51,52,53,54,55,56]. Thus, questions remain: (i) Are the different mechanisms seen in the inhibition of various types of viruses the different consequences of one molecular function of SHFL? Or (ii) is SHFL intrinsically pleiotropic? Elucidation of the molecular detail of SHFL’s antiviral function at the cell biological, biochemical, and structural levels will lead to the development of a broad-spectrum antiviral drug targeting the “Achilles’ heel” of pathogenic viruses. On the other hand, if viruses inhibit the antiviral activity of SHFL in some way, further investigation of the virus-mediated antagonizing mechanism would provide us with new insights into the molecular detail of SHFL. On top of that, although Hanners et al. reported that without ZIKV infection, SHFL KO mice were normal in comparison with WT mice, it would also be interesting to clarify the role of SHFL in disease status in non-infectious diseases such as cancer. In this context, SHFL is a fascinating molecule.

## Figures and Tables

**Figure 1 ijms-23-12619-f001:**
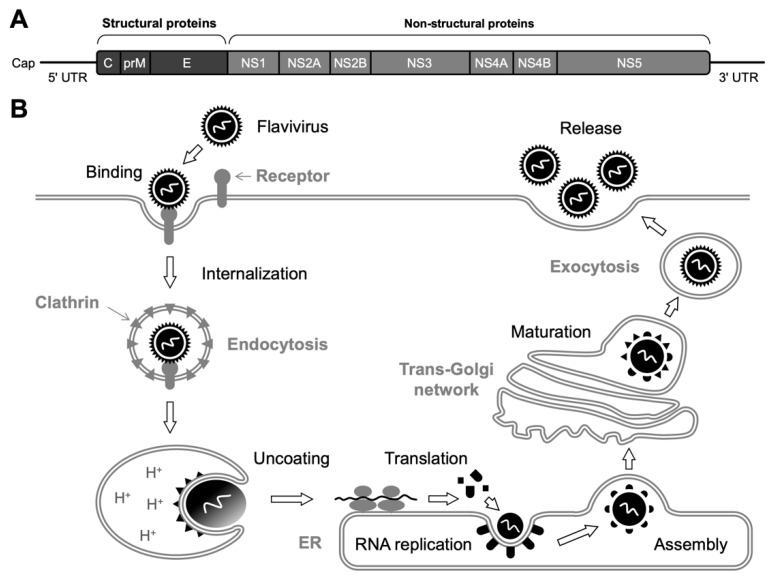
Genome structure and replication cycle of flavivirus. (**A**) Flavivirus RNA contains a single ORF flanked by 5′ and 3′ UTRs. A polyprotein precursor expressed from the single ORF is co- and post-translationally cleaved by viral and cellular proteases, yielding three structural and seven non-structural (NS) proteins. (**B**) Overview of flavivirus replication. After the binding of E glycoproteins to cell-surface entry receptors, the attached virion is internalized into the cell via clathrin-dependent endocytosis. Then, acidification of the endosomal vesicles triggers a structural rearrangement of the E protein, resulting in a fusion between the viral and endosomal membranes. A single precursor protein is translated from the released viral RNA, which is, in turn, cleaved to produce functional proteins. The viral RNA amplification process takes place on the ER membrane. Subsequently, the nucleocapsids, composed of C proteins and synthesized viral RNA, are assembled with prM and E heterodimers and cellular lipid bilayers to form immature (i.e., non-infectious) particles. However, conformational changes in the glycoproteins occur during the transport through the Golgi apparatus, and eventually, mature (i.e., infectious) virions are released by exocytosis.

**Figure 2 ijms-23-12619-f002:**
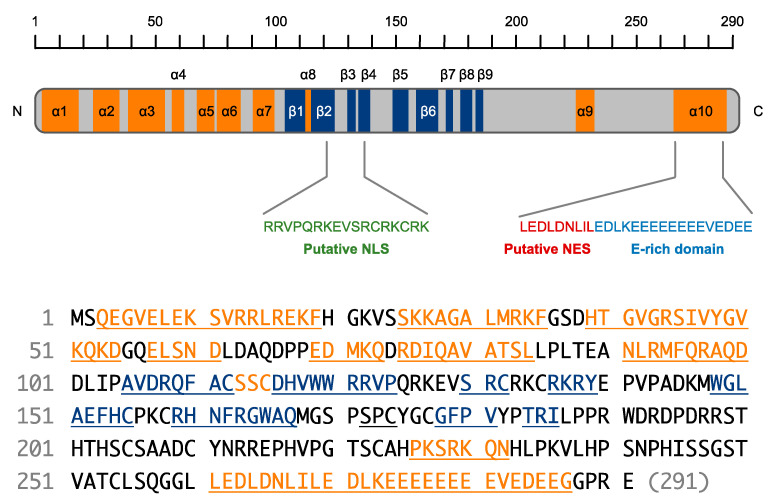
Secondary structure of SHFL. SHFL (291 amino acids) is predicted to be composed of ten α-helixes (orange) and nine β-strands (shown by navy blue). Amino acid sequence-based protein motif prediction programs also show that SHFL contains putative NLS (green, by cNLS Mapper [http://nls-mapper.iab.keio.ac.jp/cgi-bin/NLS_Mapper_form.cgi], accessed on 28 September 2022) and NES (red, by NetNES [https://services.healthtech.dtu.dk/service.php?NetNES-1.1], accessed on 28 September 2022). A characteristic E-rich domain (blue) is located next to the NES sequence. The α-helixes (orange) and β-strands (navy blue) are also indicated in the amino acid sequence of SHFL (bottom part).

**Figure 3 ijms-23-12619-f003:**
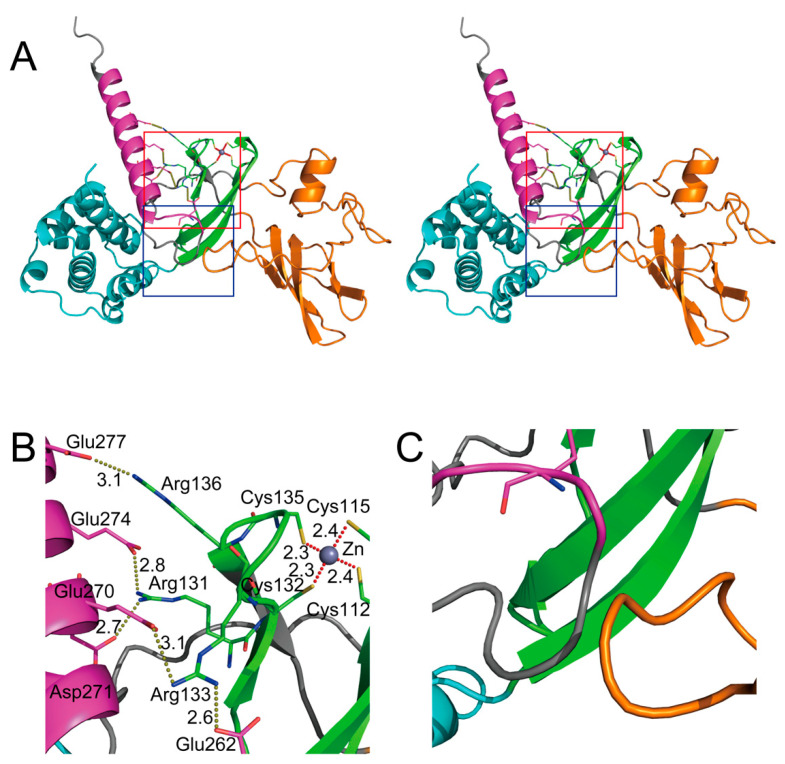
Three-dimensional structure of SHFL predicted by AlphaFold2. (**A**) Stereodiagram of the overall structure. The zinc ribbon domain, E-rich domain, helix-rich domain, and flexible domain are colored green, magenta, cyan, and orange, respectively. Red and blue squares indicate areas in Figure 3B,C, respectively. (**B**) Enlarged view of the core region. The position of the Zn^2+^ was manually adjusted to be equidistant from the S atoms of the four cysteine residues (Cys112, Cys115, Cys132, and Cys135). Brown and red dotted lines represent electrostatic interactions and coordination bonds to Zn^2+^, respectively, with distances (Å) between the atoms. (**C**) Enlarged view of the putative hydrophobic interaction between two loops.

**Table 1 ijms-23-12619-t001:** Summary of the activity of SHFL in the inhibition of flaviviruses.

Flaviviruses	Proposed Mechanisms-of-Inhibition	Notes	References
DENV, WNV	Translational inhibition of viral mRNA	Identified by a gain-of-function screen using an IFN-related cDNA library Interacting with PABPC1 and LARP1 The middle domain of SHFL containing the NLS and zinc ribbon domain accounted for the DENV inhibition	[15]
DENV	Degradation of viral mRNA	Interacting with cellular RNA-binding proteins including MOV10 Associated with the viral replication complex	[22]
ZIKV	Destabilization of viral NS3 protein	Mediated by a lysosome-dependent pathway	[23]
JEV	Inhibition of ribosomal frameshifting in NS1′ protein expression Destabilization of viral NS3 protein	Required zinc ribbon domain A shorter isoform of SHFL (lacking residues 164–199) did not exhibit the inhibitory activities	[26]
ZIKV, YFV, WNV, DENV	Blocking at a later stage of virus replication	Co-localization of SHFL with dsRNA in YFV-infected cells (in vitro) SHFK KO mice infected with ZIKV exhibited reduced survival and neuropathological outcomes The level of ZIKV replication was increased in SHFL KO mice when compared to WT mice	[27]

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
