# Peer review of "Restriction of Flaviviruses by an Interferon-Stimulated Gene SHFL/C19orf66"

_ijms, 2022, doi:10.3390/ijms232012619_

Round 1
Reviewer 1 Report
It is important to find or summarize the new factor involved in the restriction of flaviviruses. This review characterizes a new inhibitor, SHFL for blocking flaviviruses infection in vivo and in vitro. It is worth publishing.
specific comment
1. Please add the full name of -1PFR
2. In section 7. anti-flavivirus activity of SHFL in vivo, authors only included the ZIKV, how about other flaviviruses? If you could, please include DENV, JEV, and WNV.
Author Response
Responses to Reviewer 1 Comments
It is important to find or summarize the new factor involved in the restriction of flaviviruses. This review characterizes a new inhibitor, SHFL for blocking flaviviruses infection in vivo and in vitro. It is worth publishing.
We appreciate the reviewer’s positive evaluation and helpful comments. Italics refer to new locations in the revised manuscript.
Point 1: Please add the full name of -1PFR
Response: The full name of PRF (programmed ribosomal frameshifting) has been mentioned in lines 236−237. -1PRF is one of the modes of PRF, in which the ribosome slides back one nucleotide to the 5' direction at the end of one ORF so that the other ORF becomes in-frame with the first reading frame. A brief explanation of the -1PRF is also described in lines 239−242.
Point 2: In section 7. anti-flavivirus activity of SHFL in vivo, authors only included the ZIKV, how about other flaviviruses? If you could, please include DENV, JEV, and WNV.
Response: In the study by Hanners et al., in vivo antiviral activity of SHFL using KO mice was only examined for ZIKV. However, as the reviewer commented, it would be important to investigate the replication and pathogenesis of other flaviviruses in SHFL knockout mice to clarify the physiological role of SHFL.
Reviewer 2 Report
SHFL is interesting that acts as a novel antiviral factor exhibiting suppressive activity against RNA and DNA viruses.This manuscript by Youichi Suzuki and colleagues summarize that the molecular mechanisms of SHFL-mediated inhibition in flavivirus infection and discuss the molecular and structural basis of the inhibitory mechanism.If the following points can be revised, or supplemented clearly, this study will bring new ideas. The comments are as follows:
1.Line 44, ‘three other genera, Hepacivirus, Pegivirus, and Pegivirus’ should be changed to Hepacivirus, Pegivirus, and Flavivirus.
2. Line 154 ‘5. Molecular aspects of SHFL in the inhibition of flaviviruses’,In this paragraph,Youichi Suzuki et al summarized the molecular basis for inhibition of the flavivirus by SHFL,including (i) translational inhibition, (ii) RNA degradation, (iii) protein degradation, and (iv) frameshift inhibition.It is suggested to summarize the research progress of these mechanisms systematically, which will be more intuitive to show in the form of graph or table.
3.SHFL was a fascinating molecule with broad-spectrum antiviral activity,and was also identified as an antiviral ISG. On the other hand,there may also be some mechanism that antagonizes its antiviral function. If possible, please supplement or predict the possible antagonistic mechanism.
Author Response
Responses to Reviewer 2 Comments
SHFL is interesting that acts as a novel antiviral factor exhibiting suppressive activity against RNA and DNA viruses. This manuscript by Youichi Suzuki and colleagues summarize that the molecular mechanisms of SHFL-mediated inhibition in flavivirus infection and discuss the molecular and structural basis of the inhibitory mechanism. If the following points can be revised, or supplemented clearly, this study will bring new ideas. The comments are as follows:
We greatly appreciate the reviewer's insightful comments and suggestions that surely improve our manuscript. Italics refer to new locations in the revised manuscript.
Point 1: Line 44, ‘three other genera, Hepacivirus, Pegivirus, and Pegivirus’ should be changed to Hepacivirus, Pegivirus, and Flavivirus.
Response: We have corrected this sentence (line 45). Thank you.
Point 2: Line 154 ‘5. Molecular aspects of SHFL in the inhibition of flaviviruses’, In this paragraph, Youichi Suzuki et al summarized the molecular basis for inhibition of the flavivirus by SHFL, including (i) translational inhibition, (ii) RNA degradation, (iii) protein degradation, and (iv) frameshift inhibition. It is suggested to summarize the research progress of these mechanisms systematically, which will be more intuitive to show in the form of graph or table.
Response: We agree with the comment that it would be very helpful to have a summary of the key points of SHFL’s anti-flavivirus activities reported so far. Following the reviewer’s suggestion, we added a table that summarized the activity of SHFL in the inhibition of flaviviruses in the revised manuscript (pages 6−7).
Point 3: SHFL was a fascinating molecule with broad-spectrum antiviral activity, and was also identified as an antiviral ISG. On the other hand, there may also be some mechanism that antagonizes its antiviral function. If possible, please supplement or predict the possible antagonistic mechanism.
Response: There are no reports yet on the viral evasion system antagonizing SHFL activity. However, as the reviewer expected, it will be intriguing if viruses use their molecules to inhibit the antiviral activity of SHFL, and further investigation of the virus-mediated antagonizing mechanism would provide us with new insights into the molecular detail of SHFL. We added this description in the revised manuscript (lines 354−356). Meanwhile, as described in lines 230−234, it has been demonstrated that flaviviruses, including DENV, circumvent the cellular antiviral response via the antagonization of regulator molecules in IFN responses, which should indirectly prevent SHFL from exerting its antiviral activity.
Reviewer 3 Report
In this review, the authors discuss the current understanding of the anti-flavivirus mechanisms of a novel ISG, SHFL (shiftless antiviral inhibitor of ribosomal frameshifting), and the molecular basis of the inhibitory mechanism using a predicted tertiary structure of SHFL. This review is constructive for readers to understand this new anti-viral ISG.
Comments:
1. Please add a table to summarize the molecular aspects of SHFL in inhibiting different flaviviruses.
2. Please descript whether transcriptome changes in the SHFL KO cells or mice.
3. Is SHFL involved in any signaling pathway of cellular antiviral response? If so, please draw a figure to point out the role of SHFL.
Author Response
Responses to Reviewer 3 Comments
In this review, the authors discuss the current understanding of the anti-flavivirus mechanisms of a novel ISG, SHFL (shiftless antiviral inhibitor of ribosomal frameshifting), and the molecular basis of the inhibitory mechanism using a predicted tertiary structure of SHFL. This review is constructive for readers to understand this new anti-viral ISG.
The authors appreciate the supportive suggestions and careful attention to detail made by the reviewer. Italics refer to new locations in the revised manuscript.
Point 1: Please add a table to summarize the molecular aspects of SHFL in inhibiting different flaviviruses.
Response: Thank you for the suggestion. Following the reviewer’s suggestion, we added a table that summarized the molecular activity of SHFL in the inhibition of flaviviruses in the revised manuscript (pages 6−7).
Point 2: Please descript whether transcriptome changes in the SHFL KO cells or mice.
Response: The point raised by the reviewers is of great interest to us since SHFL is an RNA-binding protein and was suggested to be involved in the degradation of mRNA. Therefore, we also agree with the reviewer’s guess that the transcriptome expression profiles may be different between WT and SHFL KO cells or mice. However, so far, no studies have reported transcriptome analysis in the absence of SHFL, which is currently under investigation by us.
Point 3: Is SHFL involved in any signaling pathway of cellular antiviral response? If so, please draw a figure to point out the role of SHFL.
Response: In a previous study, we have examined the possibility that SHFL might be a component in IFN signaling. However, overexpression of SHFL did not activate the expression of IFN-β and other ISGs (including LY6E, ISG15, ISG54, and RIG-I) in human cells. Additionally, a knockdown experiment revealed that gene expressions of ISG15, ISG54, RIG-I, and IFN-β upon treatment with type-I IFN were not reduced by the depletion of endogenous SHFL in HeLa cells. From these data, we believe that SHFL is not a regulator of the IFN-mediated antiviral response. This is newly mentioned in the revised manuscript (lines 123−125).